# Integrated Impact of Circular Economy, Industry 4.0, and Lean Manufacturing on Sustainability Performance of Manufacturing Firms

**DOI:** 10.3390/ijerph20065119

**Published:** 2023-03-14

**Authors:** Ahmed M. Ghaithan, Yasser Alshammakhi, Awsan Mohammed, Khwaja Mateen Mazher

**Affiliations:** 1Construction Engineering and Management Department, King Fahd University of Petroleum & Minerals, Dhahran 31261, Saudi Arabia; 2Interdisciplinary Research Center of Smart Mobility and Logistics, King Fahd University of Petroleum and Minerals, Dhahran 31261, Saudi Arabia

**Keywords:** circular economy, industry 4.0 technologies, lean manufacturing, sustainability performance, Saudi Arabia

## Abstract

In today’s business environment, contributions made by the manufacturing sector to the economy and social development is evident. With a focus on long-term development, the manufacturing sector has adopted advanced operating strategies, such as lean manufacturing, industry 4.0, and green practices in an integrated manner. The integrated impact of circular economy, industry 4.0, and lean manufacturing on sustainability performance has not been adequately addressed and investigated. Therefore, the aim of this study is to investigate the integrated impact of circular economy, industry 4.0, and lean manufacturing on the sustainability performance of organizations in Saudi Arabia. Data were collected through a questionnaire-based survey as a primary data instrument. A total of 486 organizations have responded to the survey within the timeframe. Moreover, the structural equation modeling method is utilized for data analysis through SmartPLS tool for the developed hypotheses of the research. The findings highlight the positive impact of circular economy on the sustainability of the organizations. Furthermore, the results indicate that industry 4.0 and lean manufacturing have positive mediating impacts as enablers for the successful implementation of circular economy toward the sustainable performance of organizations in Saudi Arabia. The study finding confirms that lean manufacturing is a substantial mediating variable that is essential for the successful implementation of industry 4.0 technologies. Moreover, the study indicates the recognition and acknowledgment of companies on circular economy principles, industry 4.0 technologies, and lean manufacturing tools to achieve the desired sustainability.

## 1. Introduction

Sustainability refers to the process of satisfying existing needs without compromising the needs of future generations, emphasizing the importance of preserving the existing blessings of mother nature’s resources that are finite in nature [1]. The term ’sustainability of industrial organizations’ refers to achieving significant increments in their overall revenues, profitability, product or service development, and market share, as well as in attaining market expertise, developing a healthy working environment, and improving their environmental fingerprint. Sustainability is addressed in three dimensions, namely the social side, economic aspect, and environmental dimension. The three dimensions are called the “triple bottom line” (3BL) [2]. The objective of having a sustainable organization is the ultimate goal for manufacturing organizations due to the attractiveness of the positive impact on the environment, society, and economy. Sustainability has gained growing consideration from manufacturing companies and has been placed at a spotlight in the strategic goals of manufacturing firms. Yet, manufacturing firms are still struggling with the negative effects they generate on the environment and society. The problem lies in the cost and profit game of manufacturing businesses as it is usually the case that firms tend to choose to produce cheaper products and earn more rather than to produce expensive products with a lesser negative impact on the environment [3].

In fact, the industrial sector is recognized as one of the main sources of environmental pollution and resource depletion [4] due to a high consumption rate of energy, an over-consumption rate of resources, and a carelessness of the emitted gases to the atmosphere [5]. However, with an expected increment of the world population and associated boom in consumption rates and environmental effects, it shows that the usual business at its current existence is not leading the world into a better future in terms of sustainability [6]. Many researchers have recommended various approaches to encounter aforementioned challenges to transform the industrial sector into more eco-friendly industry. For example, circular economy is perceived as a good approach to reduce organizational conflicts between the economic prosperity and leaving a good impact on the environment, emphasizing the challenge to keep a business profitable and still leave a good environmental footprint [7]. Circular economy emphasizes on waste minimization as well as materials and resources optimization while maintaining them within the economy circle for much longer than expected; therefore, undoubtedly, the development process must be more sustainable [8]. From an economical perspective, the circular economy implementation benefits the world from a re-consumption concept which would act as a revenue source and an eliminator of accumulated waste and the destruction of society and the environment [9]. This is considered a great gain for manufacturing firms to achieve economic prosperity as they would generate new revenue streamlines from unutilized sources. On the social side, circular economy adopts valuable practices to an individual level, such as the collaborative consumption of product sharing/renting or product pooling that would achieve cost savings, in addition to eliminating the use of hazardous materials, which would improve the health and safety of society [10].

Therefore, circular economy has gained increased attention and strengthened its position as an effective tool toward ultimate sustainability [10]. Furthermore, in the literature, circular economy has been proven to be a positive impact on the profitability and feasibility of manufacturing firms, where additional values are created from the recovery of materials and assets that generate new sources of revenues. In addition, it helps organizations to cope with the desired growth by reshaping the way firms react to their surroundings and implement new business models [11]. The advancements in the industrial production systems unveiled new levels of innovation, which have led to the digitalization era of the manufacturing industry, thereby enabling the production systems to be more connected, integrated, and decentralized [12]. These advancements introduced the ideas of the fourth industrial revolution, or industry 4.0, to transform the production systems into more flexible, efficient, and sustainable ones with persistently high quality and low costs [13]. Furthermore, industry 4.0 can be defined as the technological revolution where people would be connected to their surrounding objects and would allow for a bridge of understanding between human and objects [14].

Industry 4.0 has positively impacted the sustainability of organizations as the core motivations for organizations to implement its technologies to achieve higher efficiency of the production systems and eliminate waste in the supply chain processes. Additionally, maintaining a higher quality of products would be reflected on more economical values for organizations [12,15]. On the social side, industry 4.0 has contributed to providing an outstanding working environment, which has been reflected on employee satisfaction, contributing to lower turnover rates and creating an attractive environment for excellence and growth [12]. Furthermore, industry 4.0 allows production systems to become more environmentally friendly, as the use of technologies have enabled proper alignment between supply chain stakeholders to eliminate the waste of materials, energy, and human resources leading to a positive impact on the environment [12]. Lean manufacturing principles have been proven to help manufacturers address the challenges of sustainability. Lean manufacturing principles are practices that have been implemented in many industries to accomplish operational excellence through the adoption of tools that promote waste reduction, production optimizations, and emphasis on continuous improvement [16]. Lean means eliminating the waste and maximizing the process output by modifications made to the people, equipment, or the process. Waste (“Muda” in Japanese) contains seven forms: overproduction waste, waste associated with waiting time, transportation waste, inventory waste, processing waste, waste of motion, and waste from product defects [17].

The relationship between lean manufacturing and sustainability performance has been studied and has unveiled a solid correlation as lean manufacturing tools and principles directly influence sustainability through the accomplishment and continuous improvement of waste reduction, thus leading to more efficient operations with higher revenue streamlines for the manufacturing firms [18]. Therefore, lean manufacturing approaches are vital for firms to reach the sustainability objectives [19]. However, after examining the three angles of sustainability with regard to lean manufacturing, it has been found that lean manufacturing positively impacts the environmental aspect, as it is mainly oriented to the waste elimination concept leading to proper distribution and consumption of resources, as well as minimizing the intensity of the hazardous emissions and pollutant materials [20]. Furthermore, the implementation of lean manufacturing in the production floor has helped organizations to achieve flawless executions, including cost reduction and increased profit margins, which accomplish and enhance the economic indices of manufacturing firms [21]. From a social perspective, lean manufacturing principles facilitate and organize the production flow, which improve the working environment and promote occupational health and safety [22]. 

The literature has shown a relationship between industry 4.0 and the effects of circular economy on the sustainability of organizations, revealing that the adoption of 4.0 industry technologies enhances the transition toward circular economy due to their technological capabilities to track resource consumptions and emissions. Indeed, industry 4.0 has helped organizations in prompting organizational innovation to merge both material and machinery with data resources in order to accomplish circular economy principles that would directly facilitate the achievement of sustainability objectives [23]. Therefore, industry 4.0 can incorporate the principles of the circular economy, creating a lucrative firm that is oriented on the systemic use of technologies. Industry 4.0 technologies and lean manufacturing are creating attractive platforms for manufacturing firms to improve sustainability performance. However, the implementation of the dual approaches, which consist industry 4.0 technologies and lean manufacturing, would result in a remarkable impact on all three dimensions of sustainability. For instance, the lean manufacturing tools are expected to eliminate waste through process optimization, whereas industry 4.0 would eliminate waste through utilizing data and innovative technologies, thus, the dual approach would boost the sustainability performance of the organizations. Furthermore, the relationship between industry 4.0 and lean manufacturing principles have been studied in the literature in different manufacturing applications [22,24,25]. Moreover, the literature suggests integrating industry 4.0 technologies with lean manufacturing as an innovative solution for any upraising challenges that would restrict the proper execution of lean manufacturing tools and principles through utilizing the power of technologies associated with industry 4.0 [22]. The impact of industry 4.0 technologies and lean manufacturing on sustainability performance has been studied, and it has revealed a solid relationship between industry 4.0 technologies and achieving sustainability performance. in addition, the adoption of lean manufacturing tools and principles have shown a strong relationship with sustainability performance. Furthermore, the literature has proven that lean manufacturing principles are a significant mediating variable alongside industry 4.0 in order to achieve the sustainability of manufacturing firms [26].

The literature indicates that there is a lack of research in exploring the triple impact of circular economy, industry 4.0, and lean manufacturing on the sustainability performance of manufacturing firms. To the best of our knowledge, there is no research in the literature that investigates the impact of circular economy, industry 4.0, and lean manufacturing on the sustainability performance of manufacturing firms. Thus, this study is intended to explore the integrated impact of circular economy, industry 4.0 technologies, and lean manufacturing on the sustainability performance of manufacturing firms in Saudi Arabia. Furthermore, the paper assesses the mediating impact of industry 4.0 and lean manufacturing on the relationship between circular economy and sustainability performance. Therefore, the importance of this paper is outlined in the results of the integrated impact of these dimensions and their reflection on the sustainability of manufacturing firms, which subsequently unveil facts and figures that aid in taking a holistic approach to applying circular economy, industry 4.0, and lean manufacturing tools and principles toward performance improvements and sustainability.

This study aims to benefit the current manufacturing firms, investors, and regulators on the impact of circular economy with regard to performance sustainability, as well as examine the impact of industry 4.0 technologies and lean manufacturing principles when applied alongside circular economy approaches. The study purpose is to help to take appropriate decisions to regulate current markets toward more eco-friendly systems from regulators’ perspectives, and help existing manufacturing firms by guiding them through a better understanding of growth tools that they could adopt and implement. In addition, investors may be navigated to make the right decisions for new investments in the manufacturing industry by utilizing the appropriate tools and systems that would maximize overall long-term sustainability. 

## 2. Background

### 2.1. Sustainability Performance

The primary focus of organizations is to keep their operations sustainable in order to grow and compete in the marketplace. Sustainability is achieved through many factors, including the maximization of the organization’s economic gains, enhancement of its human resources and innovative ways to maintain expertise within the organization, as well as by improving the environmental footprint of its operations. “Sustainability means that business success is determined not solely in financial terms, such as profits and return on investment, but also accounts for environmental and social dimensions” [27]. Manufacturing firms must satisfy their clients, customers, suppliers, society, and governments needs and expectations. Therefore, to achieve the ultimate sustainability, the manufacturing firms shall address three dimensions: socio, economic, and environmental, which are known as the triple bottom line of sustainability [2]. Economic sustainability performance (EP) is “vital to corporate financial success; an organization must be able to produce goods and services on a continuous basis while also making a profit in order to survive” [28]. Sustainable economic performance is a crucial indicator for the vitality of a business to operate on promising long-term perspectives. It determines numerous decision-making activities that directly or indirectly influence other organizational factors [2].

Social sustainability performance (SP) represents the humanitarian context of the organization, which “emphasizes fairness in distribution and opportunity and relates to issues of health and education, income inequality, and poverty” [29]. The organization “should serve as a place for people, focusing on collaborative learning and development of human capacities” [30]. Therefore, to strive in the marketplace, the organization has to prioritize its social responsibility to achieve an environment of growth and innovation among its employees, which is reflected on an increased sense of social belonging and loyalty, leading to overall sustainability [2]. Environmental sustainability performance (EVP) addresses the impact of the organization to the environment as a result of its routine production activities [28]. Therefore, to ensure sustainability in regard to the environment, the organization has to either aim for net zero emissions or aim to leave a positive environmental footprint in its local surroundings, thereby “improving the quality of air and water, exploiting local waste flows, providing renewable energies, and acting as storage for surplus energy” [30].

### 2.2. Circular Economy

The circular economy represents a paradigm that decouples organizational growth and resource consumption. It sets new channels of production that form a set of strategies focused on minimizing resource consumption, improving efficiency, and achieving waste reduction [31]. Furthermore, the circular economy paradigm intends to grant the circulation of resources within a closed loop, leading to the overall reduction of the need for new materials as inputs into production systems [7]. In addition, researchers and practitioners have recognized how the circular economy principles are producing entirely new and highly innovative business models, such as sharing platforms, remanufacturing, modular design, and circular supplies, which are radical and crucial approaches that profoundly transform the current market culture by adopting circular loops [6]. Furthermore, the fundamental definition of the main elements of circular business models can be derived from the essential principles of the circular economy. Numerous business models were suggested in the literature to help translate the circular economy principles into well-organized actions and responsibilities. A very well-established and defined business model is called the ReSOLVE framework, which consists six business actions to execute the principles of the circular economy (regenerate, share, optimize, loop, virtualize, exchange), and which indicates the major opportunities depicted by the ReSOLVE framework [10].

#### 2.2.1. Regenerate Criterion

Regenerate implies the concepts of converting the waste into a reusable asset that could be involved again in the production and economy cycles. The regenerate criterion emphasizes circular supplies to ensure minimal waste and resource consumption, which includes using renewable energy as the main source of production power and using biomaterials as raw materials that are recyclable, ensuring regenerating value for the economy [10]. In addition, it includes practices such as locating production activities in efficient buildings to ensure lower energy consumptions, as well as eliminating the use of hazardous materials that could negatively impact the social and environmental aspects of the world. Furthermore, regenerate is “related to returning recovered biological resources to the biosphere. Thus, it aims to reclaim, retain, and regenerate the health of ecosystems” [31].

#### 2.2.2. Share Criterion

The share criterion intends to maximize consumer access to products and services. This could be accomplished through peer-to-peer sharing of private or public products or service in order to balance consumer needs and the rate of material consumptions. Therefore, maximizing the utilization of products and services is the main goal of share value to achieve better resource distribution between users, which significantly leads to an overall reduction in the world’s overall raw material consumption. Furthermore, the essence of share could also be achieved through the reuse of old products and items that are considered as a waste and recover them back into the production cycle. In addition, the extension of a product lifecycle is a core emphasis in this clause via many practices, including maintenance, repair, refurbishment, and upgrading of products [6]. Furthermore, stretching the product lifecycle would involve actions at the pre-existing stage of the resource, such as at the design stage, to ensure durability enhancements. 

#### 2.2.3. Optimize Criterion

Optimize focuses on improving the performance and efficiency of products, as well as optimizing supply chain activities to eliminate waste and create additional value. However, this emphasis could be implemented through many applications in the industry, such as asset management, product on-demand, waste reduction, and efficient outsourcing. These practices ensure the proper usage of recycled materials and products and eliminate the waste generated from poor production planning as well as introducing solutions, such as outsourcing to ensure the efficient and proper use of goods, materials, and human resources. Furthermore, waste reduction addresses the entire value chain, and this could be accomplished by implementing lean practices. Moreover, optimizing tools include leveraging cutting-edge technologies, such as big data, automation, remote sensing, and steering. It is important to note that optimized criterion does not include the change of product or technology [10].

#### 2.2.4. Loop Criterion

Loop criterion highlights the importance of keeping the materials and consumables within a closed loop. It calls for the innovation to upscale the value gained from the production activities. However, this includes concepts such as product transformation and remanufacturing, which is the process of restoring the product to its brand-new status to attract consumers to this type of manufacturing activity. Furthermore, the upcycling principle is one of the principles which hunt for consumer satisfaction, where materials are recycled and transformed into products with upgraded values to sell to consumers. Therefore, the main goal of the loop criterion is to create new ways to ensure materials are maintained within the economy loop to achieve overall resource consumption and increase value creation [15].

#### 2.2.5. Virtualize Criterion

The virtualize approach is concerned with the dramatic shift in how consumers preserve the product or service. It introduces new and innovative ways that could provide the same experience while eliminating the need to consume the same amount of resources. Therefore, the dematerializing concept is the main focus of this criterion, where the ultimate goal is achieved when a physical product, service, or process is shifted to a virtual form while maintaining the same function delivered to the users. Practices such as free-paper use in factories is considered a good example of the virtualize principle [31]. 

#### 2.2.6. Exchange Criterion 

Exchange actions are focused on the process of the replacement of existing materials, products, and services with more efficient and eco-friendly ones. However, the exchange criterion includes replacing old materials that obstruct the recycling and remanufacturing process with more effective and efficient materials. In addition, changing production capability to achieve higher productivity and task complexity through new technologies, such as 3D printing machines, improves the production rate and enables the performance of very complicated tasks. Therefore, the exchange criterion is addressing the need to transform the process by selecting the perfect fit of materials or technologies to achieve circularity of the manufacturing [10].

### 2.3. Industry 4.0 Technologies

Industry 4.0 consists a variety of technologies that enable the advancement of digital and automated production systems as well as the digitization of the value chain [15,32]. This would lead to significant enhancements of the product quality and reduce the cycle time to release the product to the market, which would be reflected in the overall firm performance. In addition, industry 4.0 allows the communication between products, surrounding domain, and business stakeholders, which leads to proper production planning and overall improvement in the value chain [33]. Furthermore, industry4.0 implementation will transform the production factory from processes and IT support to an integrated cyber-physical production system, as data will play significant roles on how the factory manages its processes as well as input and output activities [34]. Industry 4.0 technologies could take many forms, including IoT—internet of things, big data analytics, cloud computing, additive manufacturing, robotic system, and augmented reality.

#### 2.3.1. IoT—Internet of Things (IOT)

IoT is a key technology for intelligent manufacturing systems, enabling the intelligent identification of objects, live and active location tracking, the management and monitoring of objects with a commonly used device, the interconnection of autonomous terminal and advanced international services, as well as other vital factors. Moreover, the intelligent manufacturing systems comprise the collection and generation of production equipment data, information services data, and industrial production service data of the same field. The evolvement of the internet of things into the factory allow the deployment of connected device technologies that enable human and machines to achieve efficient interactions, which help to facilitate the manufacturing process effectively [35].

#### 2.3.2. Big Data Analytics (BDA)

Big data analytics is defined as “a holistic approach to managing, processing and analyzing the 5 V data-related dimensions (i.e., volume, variety, velocity, veracity and value) to create actionable ideas for delivering sustained value, measuring performance and establishing competitive advantages” [36]. However, big data analytics would achieve organizational excellence and an ultimate productivity boom through the adoption of data examinations and interpretations to uncover hidden traits, correlations, and meaningful insights. BDA utilizes tools, technologies, and infrastructure, such as social media platforms, human interaction devices, automated identification technologies, and cloud platforms to achieve and sustain innovation, competition, and productivity. Furthermore, BDA significantly allows for enhanced decision-making activities that are derived by the analysis of a large amount of data to accomplish the desired organization, learning, and innovation processes. Therefore, BDA reformats customer relationship management and improves risk assessment and mitigation activities, as well as contributes to the better sustainability performance of manufacturing firms [37,38].

#### 2.3.3. Additive Manufacturing (AM)

Additive manufacturing is defined as a technology that enables complex task execution while maintaining lower labor hours by a digital model and 3D printing. However, additive manufacturing has many capabilities to enable bulk production due to the combination of latest technologies with data analysis, leading to overall waste reduction, rapid production, and labor reduction [39]. Additive manufacturing uses various modeling technologies, such as computer-aided design (CAD) and 3D scanning software, to achieve performance enhancements and execute complex geometries with repetitive capabilities, which reduce the overall labor hours and improve production efficiencies [25].

#### 2.3.4. Cloud Computing (CC)

The technological advancements on human interactions with surrounding objects has created massive flows of data that can cause problems in storing and processing these flows, which has eventually led to the introduction of cloud computing concepts to overcome these challenges. Cloud computing addresses the fact that large flows of data cannot be managed by current computers due to its limited processing power. Therefore, cloud computing has helped manufacturing companies to store, analyze, process, and manage data, where they are located at more than one location provided by specialized organizations, allowing manufacturing firms to focus on their core activities and leave data concerns to outsourcing alternatives [40]. Moreover, cloud computing utilizes the advantages of flexibility, storage, sharing, and easy accessibility to help organizations implement their quests to convert data flows into useful tools to improve overall performance [40].

#### 2.3.5. Robotic Systems (RS)

Robotic systems can be defined as a “set of techniques concerning the operation and use of automata (robots) in the execution of multiple tasks in place of man for how to do a thing; standard; method; system” [41]. Robotic systems aim to improve the execution of work by assigning repetitive tasks to machines automating them. The automation is implemented by utilizing the software and AI technologies that could deliver the tasks with high levels of accuracy and efficiency [41]. Robotic systems provide smart services and innovative solutions by interacting with their surrounding environment through the use of several types of sensors, actuators, and human interface systems. Robotic systems allow bulk productions to be performed and help organizations to meet the market demand with high precision and limited resources [42]. 

#### 2.3.6. Augmented Reality (AR)

Augmented reality combines virtual information and items and real-world visions in real-time activities through the use of computer technology to provide users with distinct experiences [43]. Moreover, augmented reality is an effective tool to simulate consumer experience to provide manufacturing firms with real-time information that could affect the quality and dependability of the products or services. Therefore, the implementation of augmented reality applications would largely contribute to the better alignment of the company position in the market and improve reactions to consumer concerns [44].

### 2.4. Lean Manufacturing

Lean manufacturing can be described as a multi-dimensional production technique that consists a variety of industrial principles oriented to gain the added value in the process from a customer’s point of view by enabling flawless executions with smooth movements, where all supply chain activities reach the customer with minimum or no waste [45,46]. Therefore, the primary focus of lean manufacturing is the enhancement of the production process to achieve reliability, efficiency, and capability. In recent decades, organizations have adopted lean manufacturing in many areas, resulting in significant improvements in their performance and competitiveness [47,48,49]. In this study, six lean manufacturing dimensions were selected to cover the four major lean manufacturing factors: supplier, customer, process, and control and human factors. The dimensions are supplier development, just in time, customer involvement, continuous flow, statistical process control, and employee involvement [22]. 

#### 2.4.1. Supplier Development (SD)

Supplier development is the process of improving the relationship between customers and suppliers to build mutual understanding that would be reflected in the overall performance of both customer and supplier. It is an important measure that addresses the interactions toward a continuous improvement of the supplier’s performance by evaluating supplier indices and competencies and providing practical solutions [22]. 

#### 2.4.2. Just in Time (JIT)

Just in time evaluates the engagement of suppliers to ensure that the right amount of resources is delivered at the right time and with the right quantity, thus eliminating the need to wait for missing items or the need to store items for longer than needed by the production plan. JIT is analyzed by evaluating supplier involvement in new product development and minimal variance in desired product time delivery [22]. 

#### 2.4.3. Customer Involvement (CI) 

It is crucial to evaluate the organizational performance from a customer’s perspective to overcome their concerns and achieve customer satisfaction. This principle is intended to assess the close relationship between the organization and the customer, and between the customer’s involvement in continual product improvement and the new product’s development process. The customer demand information is continuously collected and monitored [22]. 

#### 2.4.4. Continuous Flow (CF) 

The continuous flow idea is addressing the design of the production floor to ensure proper measures are placed for a smooth continuous flow, minimizing the waste resulting from major halts or downtime. CF is measured by analyzing the proper grouping of production items, the proper grouping of equipment and workstations, and factory layout [22]. 

#### 2.4.5. Statistical Process Control (SPC) 

Statistical process control (SPC) is the process of utilizing the data and statistics to monitor the performance on the production floor. SPC is a lean manufacturing concept that helps to ensure that the process produces more specification-conforming products with less rework or scrap and operates efficiently [22].

#### 2.4.6. Employee Involvement (EI) 

Human resource is a vital factor for organizational success and the engagement of employees would enable the organization to react wisely to operational challenges. Employee involvement is assessing the level of engagement with employees from the organization leadership and their problem-solving activities. The above lean manufacturing dimensions can be further grouped into major factors. 

## 3. Hypothesis Formulation 

The hypothesis formulation was developed following the literature discussions provided in Section 1 and Section 2. The integrated impact of the circular economy, industry 4.0, and lean manufacturing on the sustainability of manufacturing firms was explored, and three hypotheses were developed along with a framework, as shown in Figure 1. Furthermore, the research assessed the direct impact of circular economy models on the sustainability performance of organizations. In addition, the authors assessed the meditating effects of industry 4.0 and lean manufacturing on the relationship between circular economy and sustainability performance.

The development of these hypotheses was initiated based on the need to study the direct impact of circular economy with respect to the sustainability performance, and then study the impact of the circular economy with the presence of mediating factors, which are industry 4.0 technologies and lean manufacturing tools to examine their impact and allow to provide a clear overview on the sustainability performance with all dimensions measured and analyzed in the study. Therefore, the assessment can be implemented by formulating the following three hypotheses: 

**Hypothesis** **1** **(H1).**
*Circular economy principles directly and positively influence the sustainability performance of organizations in Saudi Arabia;*


**Hypothesis** **2** **(H2).**
*Indirect relationship between circular economy principles and sustainability performance is significantly mediated by industry 4.0 technologies;*


**Hypothesis** **3** **(H3).**
*Indirect relationship between circular economy principles and sustainability performance is significantly mediated by lean manufacturing.*


## 4. Materials and Method

### 4.1. Data Collection

Based on the above literature, a questionnaire-based survey is established and distributed among Saudi Arabian organizations to examine the impact of circular economy alongside industry 4.0 and lean manufacturing. Saudi Arabia was selected due to the rapid movements in the country toward a more diversified economy and the manufacturing sector has been the main focus to achieve this promising goal from all aspects. The scale and magnitude of work which the government is currently implementing is considered untraditional, and the decisions tend to be abnormal in nature to achieve the desired objectives. Therefore, studying the market has provided an arena of confidence to the decision makers to make challenging decisions toward sustainability. The questionnaire was formulated to cover the elements of each dimension of the research and it was pre-tested and evaluated with middle and high management personnel. The obtained feedback helped to reformulate the questionnaire to avoid any misconception or misunderstanding. 

The questionnaire was sent by emails and through social media channels. The answers were collected through specialized survey platforms. The questionnaire contained the following sections to address the research dimensions: Circular Economy Performance, industry4.0 Implementation, Lean Manufacturing Adoption, and Sustainability Performance.

The questionnaire contained descriptive statistics of the questionnaire respondents, with 48.5% having been between the ages 25 and 35, and 35.6% between 35 to 45 years, thus covering a wide spectrum of middle to high management employees. In addition, the bachelor’s degree holders dominated the percentage of respondents from a degree perspective, with higher education holders accounting for 26.3%, which provided the study with a more reliable input. Furthermore, managers made up 42.1% of the respondents, which indicates a good involvement of managerial role holders. (Figure 2 and Figure 3). 

The descriptive questions aimed to measure the specific industry of the business. The second question was intended to assess the size of the firms varying between small, medium, and large enterprises. The classifications were derived from the General Authority of Statistics—Saudi Arabia (Statistics, 2021) [50]. Then, 11 questions measured the circularity performance based on the ReSOLVE framework, which covers the six principles (regenerate, share, optimize, loop, virtualize and exchange). Furthermore, six questions addressed the implementation of industry 4.0 technologies, covering the six technological domains discussed in the literature (IoT, big data analytics, cloud computing, additive manufacturing, robotic systems and augmented reality), and a further six questions were directed to assess the lean manufacturing principles covering the four major lean manufacturing factors (supplier, customer, process, and control and human factors) [10]. Finally, 11 questions measured the sustainability performance and covered the 3BL: the economic, social, and environmental performances [2]. A total of 486 surveys were completed and gathered. As per the latest official data provided by the General Authority of Statistics—Saudi Arabia (Statistics, 2021), there is a total of 108,815 registered manufacturing firms within Saudi Arabia. The sample size for this research was calculated by substituting the below equation to calculate the derived sample size [51]: (1)$$x=Z{\left(\frac{c}{100}\right)}^{2}r\left(100-r\right)$$
(2)$$n=N\frac{x}{\left(\left(N-1\right)E^{2}+x\right)}$$
(3)$$E=\sqrt{\frac{\left(N-n\right)x}{n\left(N-1\right)}}$$

The sample size consisted 383 with a 5% margin of error and 95 confidence intervals, which is sufficient to use structural equation modeling [51]. A total of 486 surveys were completed and gathered with a response rate of 10%. The three hypotheses will be validated or rejected using structural equation modeling (SEM) SmartPLS tool which is a data analysis method used for studying the relationship between a set of dependent and independent variables within a set of constructs [52,53].

### 4.2. Normality Test 

The data arrays were filtrated and analyzed from all angles to better understand the implications of the uncovered data. Kolmogorov–Smirnov and Shapiro–Wilk were used to investigate the normality of the data set. The significance was less than 0.05 for all the variables, which revealed that the data do not meet the normality distribution due to the variance in the replies from the companies [54]. Table 1 shows the significance values for all variables for both Kolmogorov–Smirnov and Shapiro–Wilk methods. The kurtosis and skewness were other indicators of the normality of the data [55]. The kurtosis and skewness were tested for all valuables and indicated that the maximum absolute value of skewness was identified with a value of −0.478 and which was obviously within the acceptable range (an acceptable range within skewness < 2) [56]. In addition, the Kurtosis maximum absolute value was 1.169, which complies with the acceptance range (i.e., Kurtosis < 7) [56]. Therefore, the normality test highlighted various emphasis on the data with different implications, and the data was considered as not following a normal distribution. According to this conclusion, the SEM model is then derived through SmartPLS to avoid any confusion caused by the non-normality of the data.

### 4.3. Reliability Test 

A 5-point Likert scale was utilized to measure the answers of the respondents [57]. The Cronbach alpha (*α*) test was used to evaluate the reliability of the data and detect outliers in the data array. The following Cronbach’s alpha formula was used to assess the data reliability [58]:(4)$$\alpha =\left(\frac{n}{n-1}\right)\left(\frac{\bar{V}-\sum V_{i}}{\bar{V}}\right)$$
where $\bar{V}$ is the addition of variances of all points, *V_i_* is each point variance value, and *n* is the total number of points. The coefficient of Cronbach’s alpha varies on a scale from 0 to 1. The acceptable coefficient range starts from 0.7 to 1. Therefore, any indicators with scores less than 0.7 were revised and restated. The reliability test was conducted through SPSS software and derived Cronbach alpha (*α*) above 0.7 for all variables in the dataset, which indicates a perfect reliability of the data arrays. This is because, the more the value approaches one, the more reliable it becomes, as shown in Table 2. Thus, the level of reliability in the data are satisfactory. 

As a reliability test is acceptable, it means that the variables fit within each construct; hence, the SEM model can be calculated. 

## 5. Results 

### 5.1. Hypothesis Testing

The structural equation modeling (SEM) approach by SmartPLS tool was used to evaluate the hypothesis of this research. The testing process was established to analyze the direct impact of circular economy on the sustainability of the manufacturing firms. In addition, the mediating influence of industry 4.0 and lean manufacturing on circular economy led the sustainability of the manufacturing industries. The outcomes of the direct impact test would be significant or insignificant based on the *p*-value of the test, whereby, in the mediating test, the outcomes could take on one of three scenarios: the fully mediating scenario, where only the indirect influence is tested to be valid; the partial mediation scenario, which occurs when both direct and indirect influences are found to be valid; or no mediations, which take place when both direct and indirect influences are found to be invalid [52]. Therefore, two tests must be implemented to achieve the desired outcomes: the first one will assess only the direct relationship between circular economy and sustainability performance (H1); and the second test will contain all mediators to achieve an accurate result of the mediation impact and should not be implemented separately (H2 and H3).

#### 5.1.1. Circular Economy vs. Sustainability Performance

The test conducted to analyze the direct impact of circular economy on the sustainability performance of organizations in Saudi Arabia was proven to be significant with a β value of 0.674, T-statistics of 12.854, and *p*-value of 0, which indicated a high significance level of less than 0.001. Figure 4 shows the layout for the direct influence test. 

The path coefficients shown in Figure 4 explain the fit and the relationship between the constructs and their observed variables. The numbers on the arrows are T-statistics.

#### 5.1.2. Mediating Influence of Industry 4.0 and Lean Manufacturing on Circular Economy toward Sustainability Performance

The mediating test revealed a substantial impact of industry 4.0 on circular economy toward the sustainability performance of the firms in Saudi Arabia, which represents the second hypothesis (H2). For instance, the value for β was found to be 0.149. T-statistics was derived to be 3.342 and *p*-value was 0.001, which also indicates a significant correlation between industry 4.0 and circular economy toward sustainability performance. Furthermore, the third hypothesis assessed the relationship between lean manufacturing and circular economy toward sustainability performance. Likewise, the relationship was found to also be highly significant, with β = 0.333. T-statistics was found to be 5.718 and *p*-value was 0.000. Table 3 shows the summary of hypothesis tests concluded to be all significant and valid. Figure 5 shows the layout for the mediating test conducted on the SmartPLS 3.0. 

The path coefficients shown in Figure 5 explain the fit and relationship between the constructs and their observed variables. The numbers on the arrows are T-statistics. According to Falk and Miller [53], R^2^ should be at least 0.10 to be acceptable. Meanwhile, Chin [59] suggested that R^2^ of 0.67 and above is considered substantial, between 0.33 and 0.67 is considered moderate, between 0.33 and 0.19 is considered weak, and below 0.19 should be rejected. Therefore, in this study, our concern is the R^2^ for the sustainability performance, which is the dependent variable where R^2^ was 0.712, achieving substantial effect, whereas for the mediation variables, R^2^ was between 0.348 and 0.396, thus achieving moderate effect, which is within acceptable range as it is not the main goal for the mediation analysis, which aims to measure the sustainability performance with respect to independent variable (circular economy) and mediator variables (industry4.0 and lean manufacturing).

## 6. Discussion 

The manufacturing firms play an essential role in reducing unemployment and improving living conditions. Yet, the manufacturing firms appear to be lagging behind in environmental sustainability performance, which results in a heavy burden on the environment. New emerging approaches and technologies, such as circular economy, industry 4.0, and lean manufacturing, are gaining increasing interest from researchers and industries in major economies, which would assist companies in addressing and mitigating their negative impacts on the environment. The findings of this research highlight the importance of circular economy principles to achieve the sustainability of organizations in Saudi Arabia, as is consistent with the literature that implies the positive relationship between circular economy implementation and the accomplishment of the sustainability objectives in manufacturing firms [49,60].

The study revealed the desire for Saudi Arabian organizations to implement circular economy principles to pursue economic, social, and environmental sustainability. Share value was perceived as an effective circular economy tool with a high selection average, indicating a high tendency to implement this value into production processes. Exchange value as well was among the top priorities of the respondents, which indicates the willingness of the organizations to invest and transform their production processes to become more efficient and effective. Optimize criterion was also part of this agreeable zone, which highlights the desire for organizations in Saudi Arabia to continuously improve all aspects of operational performance. The study predicts a future of circularity in Saudi Arabia as manufacturing firms are at a good awareness level of the circular economy principles due to an emphasis of the low hanging fruit, which would allow organizations to maximize the gain with existing or abandoned resources, as well as the noble values that circular economy carries toward an environment with no waste and a society that could benefit from routine disposal processes. 

The analysis underlined the significant role of industry 4.0 to achieve the ultimate goal of sustainability alongside circular economy implementation in manufacturing firms in Saudi Arabia. Industry 4.0 technologies can enhance customer involvement, which would bring the voice of the customer to the early stages of production [61]. The second hypothesis, which addressed the indirect relationship between circular economy principles and sustainability performance, is significantly mediated by industry 4.0 technologies. It was found to be significant and indicated the positive relationship between circular economy and industry 4.0 technologies, since the enthusiasm of firms to achieve the outcomes gained from the implementation of industry 4.0 technologies, such as efficiency improvement, production effectiveness, planning flexibility, and reduction of complexity, represent the same motivation to adopt circular economy principles. Therefore, there are perfect correlations between the implementation of both circular economy and industry 4.0 to achieve cohesive and integrated sustainability. The study highlighted the willingness of firms in Saudi Arabia to adopt both industry 4.0 and circular economy principles to shape their organizational strategies at the same time. 

Likewise, the study concluded a strong relationship between circular economy and lean manufacturing to achieve the desired sustainability (H3), thus validating the literature indications of the positive and perfect relationship between sustainability and lean manufacturing tools [26,62]. Moreover, this study highlighted the importance of the circular economy alongside lean manufacturing principles to achieve organizational prosperity and sustainability performance. The study highlighted the vast implementation of the lean manufacturing tools and principles in Saudi Arabian manufacturing firms and highlighted the well-established environment among businesses to implement lean manufacturing. Therefore, the circular economy revolution in the future will utilize the presence of lean manufacturing tools to strengthen its emphasis as an effective approach toward sustainability. Furthermore, the implementation of lean manufacturing would help organizations that are circular economy-oriented to understand the market and link its success with other stakeholders, and such approach can guarantee the success of circular economy as a lucrative tool to achieve sustainability. The findings of this study will provide decision-makers with the means and insights into the necessity of implementing industry 4.0 technologies, circular economy, and lean manufacturing, which will jointly improve sustainability performance. This study will benefit current manufacturing firms, investors, and regulators by examining the impact of circular economy on performance and sustainability, as well as the impact of industry 4.0 technologies and lean manufacturing principles when used in conjunction with circular economy approaches. The study will assist regulators in making appropriate decisions to regulate current markets toward more eco-friendly systems, as well as existing manufacturing firms in gaining a better understanding of data analytics tools that they can adopt and implement. Furthermore, the study assists in make the best decisions for new investments in the manufacturing industry by utilizing appropriate tools and systems that maximize overall and long-term sustainability.

## 7. Conclusions 

The manufacturing firms in Saudi Arabia play a significant role in achieving a prosperous economy that is not oil-dependent. In addition, the country is reshaping the entire manufacturing industry and investing massively to attract global manufacturing players to establish solid positions in the international arena. Therefore, the aim of this study is to help shape the manufacturing industry to promote a sustainable approach that would substantially add value to the local industries to compete in the international markets. Furthermore, circular economy is an attractive principle that would significantly achieve the sustainability objectives through all of its dimensions by emphasizing innovative and lucrative tools to organizations, where the entire value chain is benefited, thus transforming the marketplace by inventing new principles of products and services, which would gain popularity among targeted markets due to the principle of the mutual gain that circular economy emphasizes and prompts. These principles help predict a brighter future for generations to follow where sustainability is the norm for all applications regardless of the manufacturing industry, which is the backbone of the international economy.

This study formulated a matrix to examine the relationship between concerned elements where it concluded a strong harmony between circular economy principles and industry 4.0 implementation objectives, leading toward sustainability goals. Furthermore, a strong harmony was found between circular economy principles and lean manufacturing strategies toward sustainability goals. Finally, a strong harmony was found between lean manufacturing strategies and industry 4.0 implementation objectives leading toward sustainability goals. The positive relationships between circular economy, industry 4.0, and lean manufacturing to accomplish the sustainable performance of firms highlighted in this study may play a good role in convincing manufacturers in Saudi Arabia to transform their operations with circular economy principles alongside the presence of industry 4.0 and lean manufacturing as enablers for the success of the organizations. 

The study assessed the industry from a wider scope and could not observe the performance and implications of this study on different sizes of organizations as the funding for making the changes into their production lines or even inventing new ways of doing business would vary based on the firm size. Therefore, study could focus only on medium and small enterprises to address the challenges restricting them from adopting circular economy principles.

## Figures and Tables

**Figure 1 ijerph-20-05119-f001:**
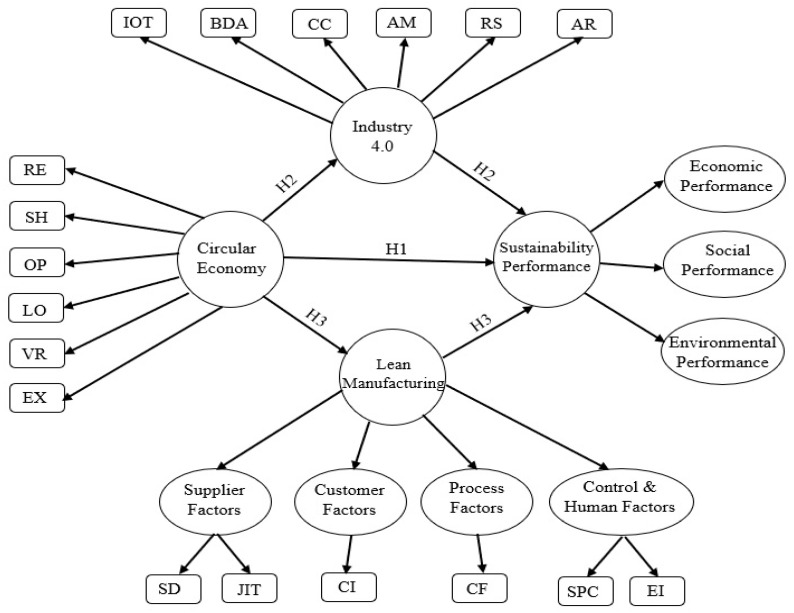
Research framework.

**Figure 2 ijerph-20-05119-f002:**
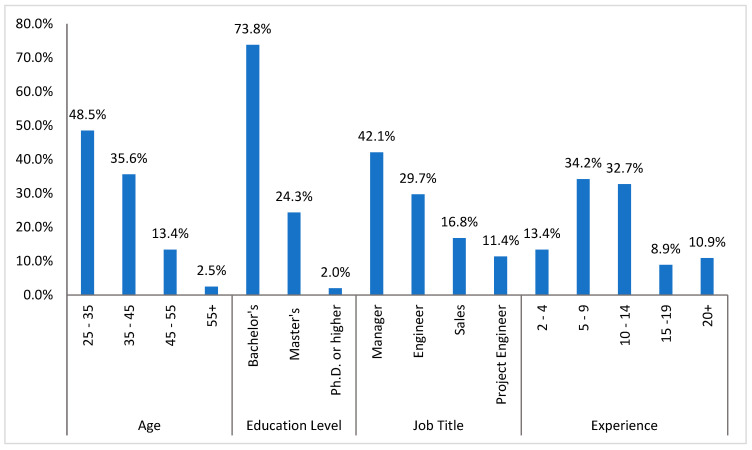
Demographic data of the respondents.

**Figure 3 ijerph-20-05119-f003:**
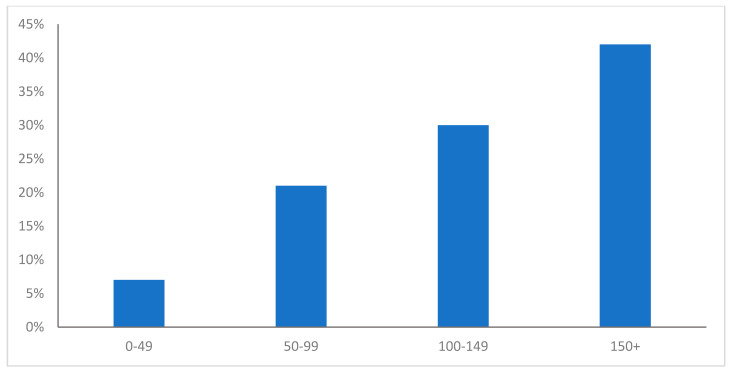
Size of the firms (number of employees) responding to the questionnaire.

**Figure 4 ijerph-20-05119-f004:**
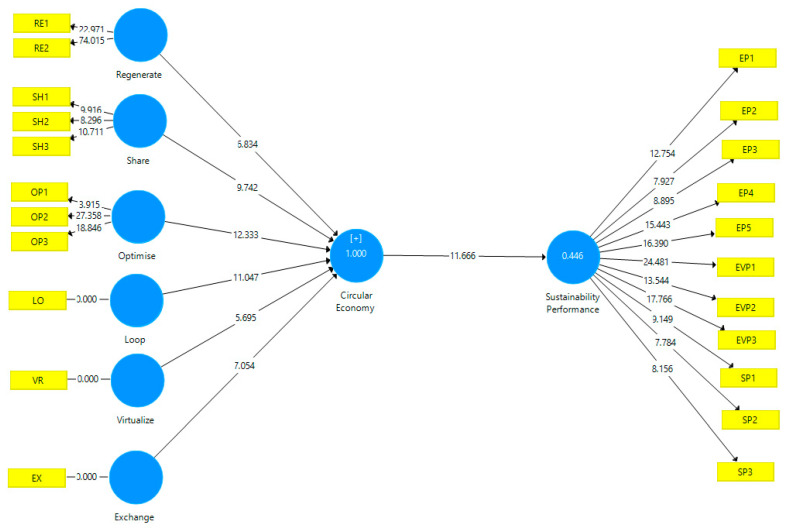
SEM direct influence model.

**Figure 5 ijerph-20-05119-f005:**
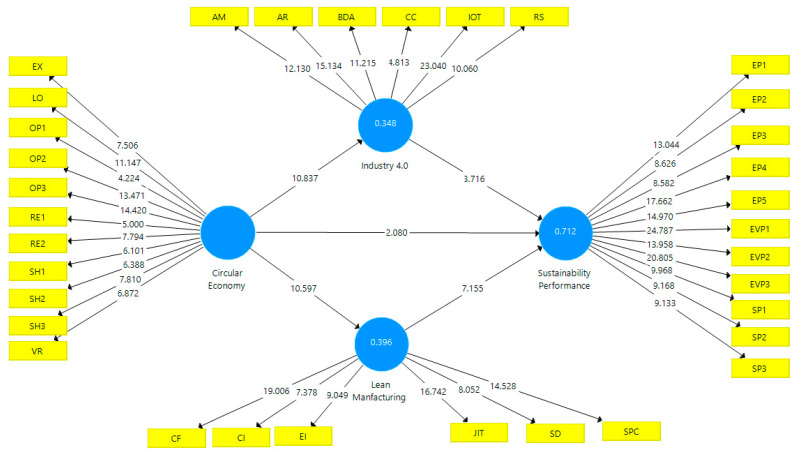
SEM-mediating influence model.

**Table 1 ijerph-20-05119-t001:** Normality test data.

Symbol	Description	Skewness	Kurtosis
RE1	Our organization has established or is planning to establish the usage of circular supplies (e.g., using renewable energy or consuming bio-based materials in the production process)	−0.875	0.340
RE2	Our organization has established or is planning to establish the consumption of recycled raw materials into production process	−0.830	−0.013
SH1	Our organization has established or is planning to establish the extension of product life cycle through maintenance and repair	−0.849	−0.089
SH2	Our organization has established or is planning to establish shared use, access, or ownership between consumers and/or businesses	−0.901	0.266
SH3	Our organization has established or is planning to establish creating value from waste and find new ways to reuse within the same firm or in a different industry/application	−0.684	−0.302
OP1	Our organization has established or is planning to establish production upon demand and secured orders	−0.619	−0.524
OP2	Our organization has established or is planning to establish waste elimination in supply chains and production processes	−0.767	−0.316
OP3	Our organization has established or is planning to establish the collection, reuse, refurbishing, and resale of used products	−1.186	0.915
LO	Our organization has established or is planning to establish restoring a product or its components to a “brand new” quality	−1.043	1.060
VR	Our organization has established or is planning to establish shifting from physical to virtual activities, services, or processes	−0.745	0.411
EX	Our organization has established or is planning to establish replacing old materials with advanced materials or adopting new production technologies	−0.762	0.026
CC	Our organization has established or is planning to establish cloud computing	−0.648	−0.197
BDA	Our organization has established or is planning to establish big data analytics (the process of analyzing large data to uncover hidden patterns and correlations)	−0.777	0.376
IOT	Our organization has established or is planning to establish internet of things	−0.667	−0.056
AM	Our organization has established or is planning to establish additive manufacturing (technology which enables the manufacturing of the most complex components through digital models and 3D printing)	−0.860	0.275
RS	Our organization has established or is planning to establish robotic systems	−0.562	−0.358
AR	Our organization has established or is planning to establish augmented reality (e.g., append virtual information to the real world to simulate consumer experience)	−0.513	−0.426
SD	Our organization established strategic alliances with suppliers and is committed to the development to gain mutual success	−0.808	0.428
JIT	Suppliers emphasized on just in time delivery to reduce delays in production flow and minimize inventory levels	−0.966	0.898
CI	Our organization established customer involvement channels for continuous improvement objectives	−1.069	1.169
CF	Our organization established mechanisms that enable and ease the continuous flow of products	−0.630	−0.393
SPC	Statistical process control is utilized on the production floor to measure process variability	−0.721	0.261
EI	Our shop floor personnel contribute significantly to problem-solving activities and drive suggestion schemes	−0.784	0.325
EP1	Our organization reduced costs of production	−0.810	0.340
EP2	Our organization improved profits	−0.810	0.585
EP3	Our organization reduced product development costs	−0.690	0.402
EP4	Our organization decreased energy consumption costs	−0.622	−0.044
EP5	Our organization reduced rejection and rework costs	−0.951	0.930
SP1	Our organization improved working environment and people’s morale	−0.685	0.405
SP2	Our organization prioritize the health and safety of employees	−0.619	0.110
SP3	Our organization improved labor relations	−0.478	−0.240
EVP1	Our organization established the reduction of solid, liquid, and energy wastes	−0.751	0.135
EVP2	Our organization established the reduction of gas emissions	−0.716	0.001
EVP3	Our organization established the reduction of hazardous material consumption	−0.622	−0.255

**Table 2 ijerph-20-05119-t002:** Cronbach’s alpha *α* tests and factor loadings.

Factor	AVE	CR	*α*	Factor	AVE	CR	*α*
RE	0.73	0.92	0.88	AR	0.77	0.91	0.7
SH	0.72	0.93	0.9	SD	0.68	0.93	0.64
OP	0.75	0.9	0.83	JIT	0.65	0.92	0.64
LO	0.74	0.94	0.93	CI	0.76	0.9	0.75
VR	0.83	0.91	0.8	CF	0.77	0.91	0.79
EX	0.73	0.91	0.87	SPC	0.72	0.89	0.73
CC	0.62	0.91	0.88	EI	0.71	0.92	0.72
BDA	0.8	0.89	0.74	EP	0.65	0.9	0.78
IOT	0.72	0.91	0.85	SP	0.68	0.91	0.69
AM	0.67	0.86	0.66	EVP	0.76	0.91	0.61
RS	0.63	0.92	0.7				

**Table 3 ijerph-20-05119-t003:** Path coefficient of the research hypothesis.

Hypothesis	Std. Beta (β)	T-Value	*p*-Value	Result
H1	0.674	12.854	0.000	Validated
H2	0.149	3.342	0.001	Full mediation exists
H3	0.333	5.718	0.000	Full mediation exists

## Data Availability

Not applicable.

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
