# Peer review of "Integrated Impact of Circular Economy, Industry 4.0, and Lean Manufacturing on Sustainability Performance of Manufacturing Firms"

_ijerph, 2023, doi:10.3390/ijerph20065119_

Round 1

Reviewer 1 Report

Abstract

Authors requested to give the introductory part of the abstract a little more conceptualisation. Why is the study necessary?

More than 400 organizations have responded to the survey 16
within the timeframe. PLEASE GIVE THE EXACT FIGURE

Structural Equation Modelling method is utilized for data analysis 17
for the developed hypotheses of the research WHICH SEM?

INTRODUCTION

Author to expand discussion on the how and who the study will be significant in the last paragraph of the Introduction

Regenerate Criterion It will be better not to put all references at the end of the long paragraph. same for share criterion 2.2.1 to 2.3.4

3 hypotheses formulation

Preferable to have  additional literature  to guide the development of the hypotheses.

The three hypotheses will be studied and measured through distributing a question- 427
naire to targeted organizations in Saudi Arabia. TIS IS NOT NECESSARY

RESEARCH METHODOLOGY

This area is weak and can be improved. It is necessary to discuss as follows and in detail

1.      Research design and context. Why Saudi Arabia, why the industry focused one. Quantitative, causal research etc

2.      Population, sampling and sample size

3.      Questionnaire development including measures

4.      Data collection and data analysis. Why SEM?

There is a problem with table 2. Where are the factor loadings?

PLEASE CHECK TABLE 3 TO ENSURE THAT YOU HAVE MEASRED FULL MEDIATION CORRECTLY. VAF, INDIRECT EFFECTS DO NOT SEEM TO COME OUT

THE DISCUSSION IS WEAK. FIRST FINDINGS. 2 WHAT DO THE FINDINGS INDICATE 3. CONSISTENCY OR INCONSISTENCY WITH LITERATURE. ADD ADDITIONAL LITERATURE

CONCLUSION IS WEAK AS AT THIS STAGE (I UNDERSTAND THAT THE AUTHORS ARE FOLLOWING THE TEMPLATE.

PLEASE CREATE SECTIONS FOR . 1 THEORETICAL IMPLICATION. 2 PRACTICAL IMPLICATIONS. 3 LIMITATIONS AND AREAS FOR FURTHER STUDY

Reviewer 2 Report

The article is about an exploratory study on the combination of a few parameters including circular economy, industry 4.0, and lean manufacturing on sustainability performance using SEM analysis methodology. Overall are good but Turnitin's results on the similarity analysis show that the current content is similar by at least 77% to other publications. Please improve it and make sure less than 25% is similar. 

The followings are additional suggestions/ comments:

Pg. 1 line 12. The statement "has not been investigated" is not appropriate because most of the citation/ references are not the latest and may not cover all publications on the other platform. 

Pg. 1. Line 30- 35. Should be deleted.

Pg 9. Line 412. Should use the past tense because the study is already performed. 

Pg. 10. 440-444. Not well written about the respondent's background and why only the group was involved in the study. 

Others:

1. Table and figure should be re-arranged and organized with proper formatting. 

2. References - Please update the latest especially the latest 5 years to ensure that your research findings are still valid and trending as popular research demands. 

Reviewer 3 Report

After reading your research document, it is not clear to me what is the originality of your proposal, as several different papers have been published in the past, dealing with the interrelationship between the presented topics, i.e. :

1.     Khan, I. S., Ahmad, M. O., & Majava, J. (2021). Industry 4.0 and Sustainable Development: A Systematic Mapping of Triple Bottom Line, Circular Economy and Sustainable Business Models Perspectives. Journal of Cleaner Production, 126655. Available at: https://www.sciencedirect.com/science/article/abs/pii/S0959652621008751

 2.     Integration between Lean, Six Sigma and Industry 4.0 technologies.   International Journal of Six Sigma and Competitive Advantage · June 2021

 3.     Industry 4.0 to Accelerate the Circular Economy: A Case Study of Electric Scooter Sharing. Sustainability 2019, 11, 6661; doi:10.3390/su11236661

 4.     Industry 4.0 technology and circular economy practices: business management strategies for environmental sustainability.  Environmental Science and Pollution Research https://doi.org/10.1007/s11356-022-19081-6

 5.     Evaluation of the Relation between Lean Manufacturing, Industry 4.0, and Sustainability. Sustainability 2019, 11, 1439; doi:10.3390/su11051439

A comparison between your study and those other - that have been published before - would help the reader to see the originality of your work.

Round 2

Reviewer 1 Report

None. I am satisfied with the corrections

Reviewer 2 Report

The updated version is a lot of improvements and is adequate for publication. Well done!

Line 524. typo error for smartPLS. please update. 
